# RELATIONBOOTH: TOWARDS RELATION-AWARE CUSTOMIZED OBJECT GENERATION

## ABSTRACT

Customized image generation is crucial for delivering personalized content based on user-provided image prompts, aligning large-scale text-to-image diffusion models with individual needs. However, existing models often overlook the relationships between customized objects in generated images. Instead, this work addresses that gap by focusing on relation-aware customized image generation, which aims to preserve the identities from image prompts while maintaining the predicate relations described in text prompts. Specifically, we introduce RelationBooth, a framework that disentangles identity and relation learning through a well-curated dataset. Our training data consists of relation-specific images, independent object images containing identity information, and text prompts to guide relation generation. Then, we propose two key modules to tackle the two main challenges—generating accurate and natural relations, especially when significant pose adjustments are required, and avoiding object confusion in cases of overlap. First, we introduce a keypoint matching loss that effectively guides the model in adjusting object poses closely tied to their relationships. Second, we incorporate local features from the image prompts to better distinguish between objects, preventing confusion in overlapping cases. Extensive results on three benchmarks demonstrate the superiority of RelationBooth in generating precise relations while preserving object identities across a diverse set of objects and relations. The source code and trained models will be made available to the public.

## 1 INTRODUCTION

Driven by large-scale text-to-image diffusion models (Rombach et al., 2022; Saharia et al., 2022; Podell et al., 2023; Pernias et al., 2023), customized image generation has recently made significant strides (Ruiz et al., 2023; Kumari et al., 2023; Wei et al., 2023; Zhang et al., 2024; Wang et al., 2024; Kong et al., 2024). This task focuses on generating images that preserve the identity of objects from user-provided inputs, enabling the creation of personalized and meaningful content. It has shown value in numerous applications, including personalized artwork, branding, virtual fashion try-ons, social media content creation, augmented reality experiences, and more.

Despite the success of many methods for customizing single or multiple objects (Liu et al., 2023; Ye et al., 2023; Li et al., 2023; Patel et al., 2024; Wang et al., 2024; Gu et al., 2024; Lin et al., 2024; Pang et al., 2024), they often overlook the relationships between objects and the corresponding text prompts. For instance, when two user-provided objects are paired with a text prompt specifying a particular relationship, the generated output should not only preserve their identities but also accurately reflect the intended relationship, such as a 'hug'. This introduces new challenges in what we refer to as relation-aware customized image generation, which focuses on preserving multiple identities while adhering to relationship prompts.

An intuitive solution to this issue is to adapt existing tuning-based or training-based methods for customizing relationships (Huang et al., 2023; mengmeng Ge et al., 2024; Materzynska et al., 2023) among objects. However, both approaches face challenges. Tuning-based methods struggle to preserve multiple identities, as they invert objects into specific tokens, while training-based schemes often fail to balance image and text prompts, frequently overlooking key textual elements and hindering the generation of relationships between objects. As illustrated in Fig. 1, previous methods fail to capture the actions described in the text prompt and lose identity preservation.

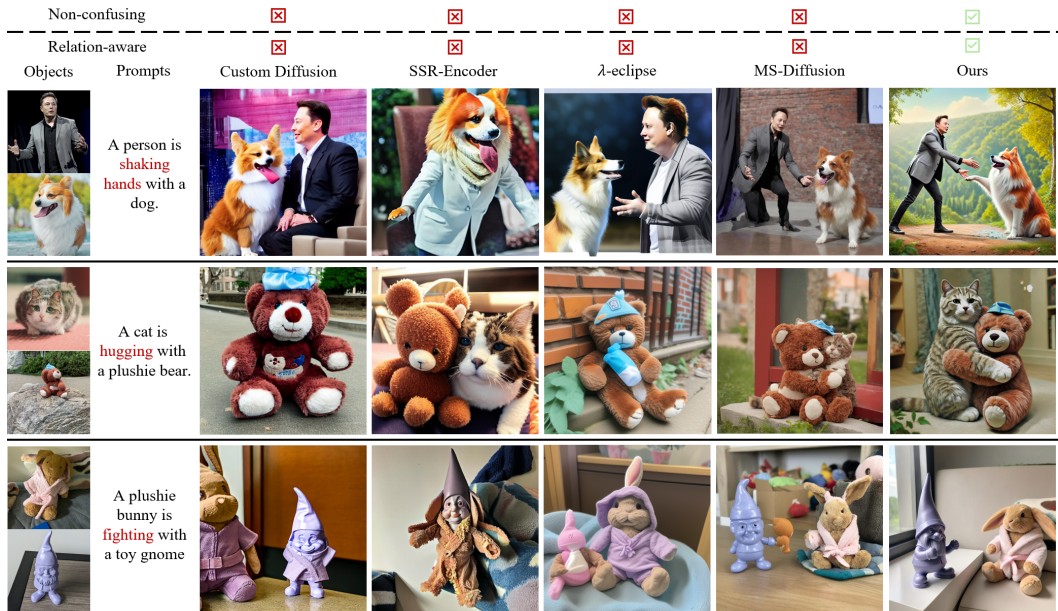

Figure 1: Relation-aware image customization, the generated image should strictly keep the predicate relation and preserve each identity among those identities provided by the text prompt.

We attribute this failure to two key factors: a lack of relevant data and an ineffective model design. Unlike data augmentation techniques such as flipping or rotation, commonly used to create paired training data in object customization methods (Yang et al., 2023; Chen et al., 2024b), our approach requires a triplet of images: two image prompts and one target image. The image prompts should contain similar objects but exhibit distinct actions compared to the target image. To collect these triplets, we propose a data engine to curate our fine-tuning set. We leverage an advanced text-to-image generation model (Betker et al., 2023) to generate triplets where the same object pair is shared across the images. Through text prompt guidance, the object in the image prompt provides strong identity information while maintaining distinct actions in the target image, enabling the model to focus on relation learning during fine-tuning.

For the model design, we propose RelationBooth, which applies the Low-Rank Adaptation (LoRA) (Hu et al., 2022) strategy to the text cross-attention layers of existing diffusion models to process user-provided text prompts. In RelationBooth, two key modules are introduced during training to enhance the customization of relationships and identity preservation. First, we introduce a keypoint matching loss (KML) as additional supervision to explicitly encourage the model to adjust object poses, since relationships between objects are closely tied to their poses. Importantly, the KML operates on the latent representation rather than the original image space, aligning with the default diffusion loss. Second, we inject local tokens for multiple objects to improve the distinctiveness of highly overlapping objects. Specifically, we employ the self-distillation method from CLIPSelf (Wu et al., 2024) to enhance region-language alignment in CLIP's dense features. Through partitioning and pooling, the well-aligned local tokens help mitigate appearance confusion between objects.

To more comprehensively evaluate relation-aware customized image generation, we constructed our RelationBench based on three established benchmarks (Ruiz et al., 2023; Kumari et al., 2023). Our methods demonstrate strong performance compared to existing approaches, achieving significant improvements in visual quality and quantitative results.

The contributions of this work are:

- We explore a novel task called relation-aware customized image generation, which aims to preserve multiple identities from image prompts while adhering to the relationships specified in text prompts. This task can enhance various user-driven applications by enabling more control and customization.

- We introduce a data engine that uses an advanced text-to-image generation model to generate triplet images where the same object pair is present with distinct actions. This well-curated dataset enables the model to focus on learning object relations during fine-tuning, leveraging the robust identity information provided by the image prompts.

- Our proposed RelationBooth method has two key modules, including keypoint matching loss and object-related local tokens, to enhance the customization of relationships and identity preservation. The extensive experiments on three benchmarks demonstrate the effectiveness of our method.

## 2 RELATED WORK

**Diffusion-Based Text-to-Image Generation.** This direction aims to produce high-quality images based on user-provided textual prompts. Diffusion-based generation models (Rombach et al., 2022; Saharia et al., 2022) encode these prompts through the text model (Radford et al., 2021; Chung et al., 2022) and inject the text embedding into the U-Net's cross-attention layers. Some methods (Podell et al., 2023; Pernias et al., 2023; Ren et al., 2024) upscale diffusion models and incorporate additional conditions as priors to generate high-resolution images. Meanwhile, several methods (Betker et al., 2023; Lian et al., 2023) introduce stronger text encoders or large language models (LLMs) to enhance the textual comprehension capabilities of diffusion models. Recent works (Peebles & Xie, 2023; Chen et al., 2024a) have replaced the U-Net denoiser with a Transformer-based denoiser. However, these models primarily focus on textual conditions and struggle to handle other forms of input, such as user-provided images. In contrast, our work facilitates customized generation while following predicate relation conditions given by text inputs.

**Diffusion-Based Customized Generation.** Customized image generation aims to produce diverse images based on user-provided concepts. Tuning-based methods accomplish this by fine-tuning specific parameters of diffusion models, thereby incorporating new concepts into text-to-image diffusion models like Stable Diffusion. Some of these methods (Gal et al., 2023; Voynov et al., 2023; Liu et al., 2023) employ text embeddings to represent customized concepts, which are then injected into prompts during inference. On the other hand, Dreambooth (Ruiz et al., 2023) fine-tunes the entire U-Net and introduces a prior preservation loss to mitigate language drift. Additionally, several approaches (Kumari et al., 2023; Gu et al., 2024; Lin et al., 2024; Pang et al., 2024) combine text embeddings and diffusion models for image synthesis. They fine-tune both the text embedding and the U-Net, yielding impressive results. Considering the substantial cost of fine-tuning for commercial uses, training-based methods (Li et al., 2023; Wei et al., 2023; Chen et al., 2024b; Patel et al., 2024; Zhang et al., 2024) have been proposed. This approach primarily utilizes an ID encoder to extract the concept's identity and inject it into the U-Net via cross-attention layers. For instance, MS-Diffusion (Wang et al., 2024) integrates grounding tokens with a feature resampler to maintain detail fidelity and employs layout guidance to explicitly place the concepts. Despite their efficiency during inference, training-based methods often struggle with identity preservation and textual condition following ability. A typical problem is the relation expressions in prompts are often overlooked. To address this issue, we present RealtionBooth, to tackle relation-aware customized generation.

**Relation-Aware Text-to-Image Generation.** Inspired by tuning-based customization methods (Gal et al., 2023), some recent works (Huang et al., 2023; Wei et al., 2024; mengmeng Ge et al., 2024) represent a "neglected word" by learnable parameters. These methods fine-tune part of the parameters on content co-existing images. For instance, Reversion (Huang et al., 2023) fine-tunes the text embedding on a set of co-existing related images and introduces a relation-steering contrastive loss. Reversion gains a better alignment between relational words and generated images by injecting new text embedding into prompts. While effective with prepositions and adjectives, Reversion struggles with predicate relation which involves significant overlaps, and cannot customize user-provided concepts. Moreover, this method still struggles to generate vivid relationships while maintaining the fidelity of multiple customized concepts, likely due to the inherent disregard for text embeddings in customization methods. Our work bridges the gap in the predicate relation-following capability of training-based customization methods. This enables efficient inference that naturally and accurately generates predicate relations from the textual prompt.

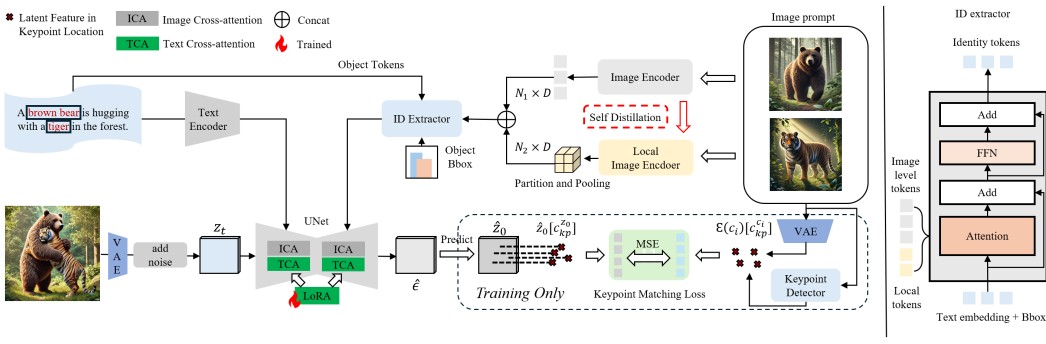

a) RelationBooth Pipeline

b) Details of ID Extractor

Figure 2: The overview of RelationBooth. RelationBooth encodes the image prompts by dual encoder architecture. After getting the U-Net output $\hat{\epsilon}$, we predicate $\hat{z}_0$ and calculate the keypoint matching loss. The part in the dotted box is only for training.

# 3 RELATION-AWARE CUSTOMIZED OBEJCT GENERATION

## 3.1 PROBLEM DEFINITION

Different from convention customization tasks, we explore a new setting that focuses on image generation by both image prompts $c_i \in \mathcal{R}^{N \times H \times W}$ and text prompts $c_t$. The $N$, $H$, and $W$ are the number, height, and width of image prompts. We call this setting RelationBooth due to the requirements that the generated image $\hat{x}$ should strictly preserve each identity and keep the predicate relation among those identities provided by $c_i$ and $c_t$.

We define this task as:

$$\hat{x} = \Phi_\theta(c_i, c_t) \tag{1}$$

where $\Phi$ is the network parameered by $\theta$. For brevity clarification, we set $N = 2$ due to the basic triplet element defined in the RelationBooth.

## 3.2 DISCUSSION AND MOTIVATION OF RELATIONBOOTH

Although previous works (Kumari et al., 2023; Gu et al., 2024; Zhang et al., 2024) have attempted to customize multiple objects under text control, we are the first to address the generation of relation-aware customized images. Notably, the state-of-the-art multi-object customization model, MS-Diffusion (Wang et al., 2024), cannot handle this task effectively. Specifically, given the image prompt $c_i$ and text prompt $c_t$. MS-Diffusion utilizes the CLIP (Radford et al., 2021) to extract the image and text tokens from $c_i$ and $c_t$ respectively. The image and text tokens are injected into the U-Net $\epsilon_\theta$ through parallel cross-attention layers. The U-Net is optimized on the latent representation of image $z$:

$$\mathcal{L}_{MS} = \mathbb{E}_{z_t, t, \epsilon, c}[\|\epsilon_t - \epsilon_\theta(z_t, t, c_i, c_t)\|_2^2] \tag{2}$$

where $\epsilon_t \sim \mathcal{N}(0, 1)$, we omit the CLIP encoder for brevity. However, despite MS-Diffusion can customize objects under text control, it's struggled for MS-Diffusion to generate relations accurately. Furthermore, MS-Diffusion might cause object confusion when heavy overlapping exists, leading to failure relation generation.

We revise the relation generation task. Existing relation inversion methods (Huang et al., 2023; mengmeng Ge et al., 2024) aim to invert a relation to a text embedding $R^*$ in the pre-trained text-to-image diffusion models. Given a set of relation-specific images $\{x_k\}_{k=1}^n$, they employ denoising loss $\mathcal{L}_{RI}$ to fine-tune $R^*$ for alignment with specific relation:

$$\mathcal{L}_{RI} = \mathbb{E}_{z_t, t, \epsilon, c}[\|\epsilon_t - \epsilon_\theta(z_t, t, c_t)\|_2^2] \tag{3}$$

Directly implementing relation inversion methods on MS-Diffusion leads to suboptimal results. We fine-tune MS-Diffusion on a set of relation-specific images, and experiment results show the model falls short in generating relation in Fig. 4. We think the image prompt $c_i$ that is not considered in

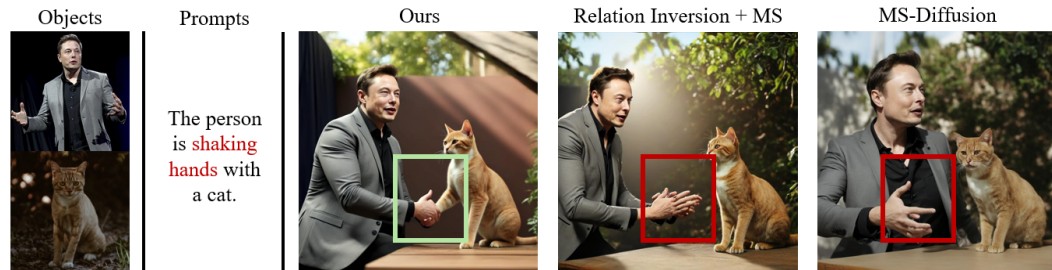

Figure 3: The overview of the data generation process. DALLE-3 can generate similar objects by using 'The photo of the same' in the text prompt on common animal categories.

| Objects | Prompts | Ours | Relation Inversion + MS | MS-Diffusion |

Figure 4: Comparison of different fine-tuning data. Simply implementing relation inversion methods on MS-Diffusion leads to failure relation generation.

relation inversion methods affects the generation process, leading to an overlook of $R^*$. In the following subsections, we will introduce the data collection and proposed method, considering lacking both of them in the proposed tasks.

### 3.3 DATA COLLECTION

Considering the scarcity of available datasets that can be used, we first design a data engine for collecting high-quality tuning data. We denote the relation-specific image as $x$. While $x$ contains relation information, it also includes the object's identities. Directly fine-tuning on a set of $x$ leads to unsatisfied relation generation. To disentangle the relation and identity in $x$. The ideal tuning data should be in the form of $\mathcal{D} = (x, c_i, c_t)$. Where $c_i$ represents the image prompts that contain the object in $x$, $c_t$ represents the text prompt that guides the relation generation. Unfortunately, directly cropping the objects from $x$ to obtain $c_i$ results in a copy-and-paste effect, even when using data augmentation techniques like flipping and rotation. Therefore, we introduce our data engine below.

**Data engine.** Fig. 3 illustrates our data generation process. Inspired by the capability of DALL-E 3 and its multi-turn dialog capability. We use DALL-E 3 to generate $\mathcal{D} = (x, c_i, c_t)$. Where $x$ and $c_i$ share the same object identity by leveraging the prompt "The photo of the same." We observe that prompting in this manner allows DALL-E 3 to remember and preserve identity in common categories such as "tiger" and "brown bear" which are enough for relation learning. Next, we employ X-Pose (Yang et al., 2024), SAM (Kirillov et al., 2023), and LLaVA (Liu et al., 2024) to annotate $x, c_i$ with keypoints, masks, and captions.

### 3.4 RELATIONBOOTH

#### 3.4.1 RELATIONLORA FINE-TUNING

To disentangle the relation and identity information in relation-specific images $x$, we utilize off-the-shelf identity inject methods from state-of-the-art customized model MS-Diffusion (Wang et al., 2024). Specifically, we utilize parallel cross-attention layers to process the $c_i$ and $c_t$, the identity information in $c_i$ facilitates the model to focus on relation information in $x$ and $c_t$ during fine-tuning. We formulate the parallel cross-attention layers:

$$h = \gamma \cdot \text{Softmax}(\frac{QK_i}{\sqrt{d}})V_i + \text{Softmax}(\frac{QK_t}{\sqrt{d}})V_t \tag{4}$$

where $Q = W_q h$, $K_i = W_{k_i} c_i$, $V_i = W_{v_i} c_i$, $K_t = W_{k_t} c_t$, $V_t = W_{v_t} c_t$, and $\gamma$ is a hyperparameter for scaling features from $c_i$. For clarity, we omit the final linear layer $W_{out}$. The predicate relation information mainly depends on text prompts. Therefore, we inject LoRA layers in all of the $W_q$, $W_{k_t}$, $W_{v_t}$, and $W_{out}$ to encourage the model to pay more attention to the predicate relation in $c_t$. We freeze all other parameters during fine-tuning. The required additional storage space for LoRA weights is only 13MB. As shown in Fig.4, benefited from the disentangled identity information in $c_i$, the LoRA weights effectively capture the predicate relation in $x$ and $c_t$, accurately generating the corresponding images. Notably, after fine-tuning, our RelationLoRA can also be directly integrated into SDXL (Podell et al., 2023), effectively enabling the model to address the relation inversion task. The extensive results are presented in the appendix (Fig. 15).

### 3.4.2 KEYPOINT MATCHING LOSS

Most relations have specific requirements for the pose, such as "hugging" requires the arm to crossed over the other object's body, and "riding a bicycle" requires the feet of the object to be on the pedal plate. Therefore, it's important to accurately manipulate the pose of objects in the right status. Intuitively, we propose to introduce keypoint features into the fine-tuning procedure to explicitly guide the pose of the objects for better relation generation.

Specifically, considering our task involves common objects instead of humans only, We employ X-Pose (Yang et al., 2024) as keypoint detector because it can detect any keypoints in complex real-world scenarios. We detect 17 keypoints for each object in $x$ and $c_i$, and denote the keypoint coordinates as $c_{kp}^x, c_{kp}^{c_i} \in \mathcal{R}^{17 \times 2}$ respectively. To encourage the model to generate accurate pose in $\mathcal{D}(\hat{z}_0)$, where $\mathcal{D}$ is the VAE decoder. We use the U-Net's output $\hat{\epsilon}$ to predict $z_0$ during fine-tuning:

$$\hat{z}_0 = \frac{z_t - \sqrt{1 - \bar{\alpha}_t}\hat{\epsilon}}{\sqrt{\bar{\alpha}_t}} \tag{5}$$

We use the VAE encoder $\mathcal{E}$ to obtain latent representation of image prompts $\mathcal{E}(c_i)$. Then we calculate the MSE loss on the corresponding keypoint locations between $\mathcal{E}(c_i)$ and $\hat{z}_0$:

$$\mathcal{L}_{KML} = \frac{1}{n_{kp}} \mathbb{E}_{z_t, c_i} \left\| \mathcal{E}(c_i)[c_{kp}^{c_i}] - \hat{z}_0[c_{kp}^{z_0}] \right\|_2^2 \tag{6}$$

$$\mathcal{L} = \mathcal{L}_{denoise} + \lambda \cdot \mathcal{L}_{KML} \tag{7}$$

where $n_{kp}$ denote the number of keypoint, $c_{kp}^{c_i}$ and $c_{kp}^{z_0}$ are keypoint's coordinates in $c_i$ and $z_0$ respectively, $\lambda$ controls for the relative weight of KML. The part inside the dotted box in Fig. 2 illustrates the model fine-tuning with the KML. We find KML is effective in encouraging the model to manipulate the pose for relation generation.

### 3.4.3 LOCAL TOKEN INJECTION

When generating relations, overlapping objects often occur. It is challenging to distinguish multiple objects solely using CLIP's image-level embedding of the full image, which can lead to confusion between customized objects during relation generation. To address this limitation, we introduce a local region representation of images. Considering the domain shift from the full image to local image regions, we employ CLIPSelf's self-distillation methods (Wu et al., 2024) to enhance the region-language alignment of local region representation. Specifically, in line with CLIPSelf, we extract dense features (Zhou et al., 2022) from a student CLIP model and derive local tokens through partitioning and pooling (He et al., 2017). We partition the full image into patches and feed them into a frozen CLIP model to obtain teacher tokens. Our approach minimizes the cosine similarity between local tokens and teacher tokens to improve the region-language alignment of local tokens. To formulate this process, we denote the hidden states of the second-to-last layer of CLIP as $h'_{clip}$. We modify the last layer of CLIP $h_{clip} = \text{ModifiedAttention}(h_{clip})$ to obtain the dense features:

$$h_{tmp} = \text{Proj}_v(\text{norm}(h'_{clip})), h_{tmp} = h'_{clip} + \text{Proj}_{out}(h_{tmp}), h_{tmp} = h_{tmp} + \text{FFN}(h_{tmp}) \tag{8}$$

where Proj means the linear projection. We discard the class embedding $h_{clip}[0]$ and reshape the image embeddings $z'[1 : h \times w]$ into an $h \times w$ feature map. We partition the feature map into patches, from which we can get local tokens $tok_l$ by pooling. During inference, we simply concatenate the aligned local tokens with the image-level tokens and send them to the ID extractor which is a

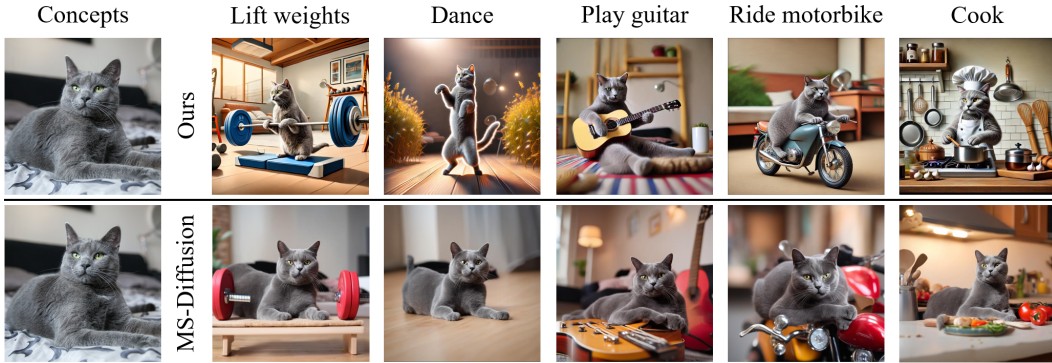

Figure 5: Single-object comparison with our base model MS-Diffusion: The results demonstrate that our method generates more accurate and natural relation-aware images.

Table 1: Quantitative comparison on RelationBench, Bold and underline represent the highest and second-highest metrics.

| Method | Single-object | | | | Multi-object | | | |
|---|---|---|---|---|---|---|---|---|
| | CLIP-T | CLIP-R | CLIP-I | DINO | CLIP-T | CLIP-R | CLIP-I | DINO |
| DreamBooth (SDXL) | 27.8 | 18.2 | 74.2 | 62.6 | 24.3 | 16.2 | 67.8 | 57.2 |
| Custom Diffusion | 26.5 | 15.2 | 73.2 | 58.8 | 20.1 | 15.4 | 64.7 | 55.3 |
| Cones-V2 | 24.4 | 13.5 | 72.1 | 57.2 | 21.3 | 15.2 | 64.3 | 54.2 |
| ELITE | 25.7 | 14.9 | 75.4 | 61.5 | — | — | — | — |
| AnyDoor | 24.5 | 14.7 | 77.4 | 62.2 | 21.6 | 14.9 | 69.7 | 59.8 |
| BLIP-Diffusion | 26.2 | 15.7 | 77.4 | 57.7 | — | — | — | — |
| SSR-Encoder | 25.5 | 15.9 | **80.4** | 59.4 | 24.2 | 14.6 | 72.1 | 56.2 |
| MS-Diffusion | 26.5 | 18.8 | 78.7 | **64.5** | 26.9 | 18.9 | 73.8 | 58.8 |
| Ours | **30.6** | **21.4** | 77.9 | 63.4 | **28.9** | **20.4** | **75.4** | **62.1** |

transformer-based architecture shown in Fig. 2 b). The query $q$ is obtained by concatenating class text embedding with the bounding box.

$$q = q + \text{GatedSelfAttn}([q, tok_i, tok_l]) \tag{9}$$

Experimental results show that local token injection enhances identity preservation in evaluation metrics and visual effects as shown in Fig. 9.

## 4 EXPERIMENTS

**Implementation Details.** For relation-aware fine-tuning, we generate a tuning set with 4-6 samples per relation. We incorporate LoRA layers into all text cross-attention layers of the U-Net. We set the LoRA rank to $r = 4$, the parallel cross-attention scaling factor to $\gamma = 0.6$, and the keypoint matching loss weight to $\lambda = 1e - 3$. The model is fine-tuned for 500 steps, using 2 A100 GPUs, with a total batch size of 8, completing the process in 10 minutes. We use the Adam optimizer with a learning rate of $1e - 4$ and no weight decay, resulting in a total of 3.1M trainable parameters. Note that our RelationLoRA only modifies the text cross-attention layers in SDXL, making it compatible with any SDXL-based models. Considering the strong identity preservation capabilities of MS-Diffusion, we implement our RelationBooth on MS-Diffusion. During fine-tuning and inference, we concatenate local tokens with image tokens. We put more self-distillation details of the Local Image Encoder in appendix (A.1). Additionally, as shown in appendix (Fig. 15), our RelationLoRA is compatible with SDXL for addressing the Relation Inversion task (Huang et al., 2023).

**Evaluation.** To evaluate relation-aware customized image generation, we propose RelationBench, consisting of 44 objects from DreamBench (Ruiz et al., 2023) and CustomConcept101 (Kumari et al., 2023), along with 25 predicate relations. The object categories include pets, plushies, toys, people, and cartoons. Using GPT-4, we generate 100 cases for single- and multi-object evaluations. Following previous works (Ruiz et al., 2023; Wang et al., 2024), we evaluate our method on three metrics: (1) Identity Preservation, which assesses the similarity between the generated images and the image prompts, using the CLIP image score and DINO score that calculate the cosine similarity between the class embeddings of images, referred to as CLIP-I and DINO, respectively; (2) Text

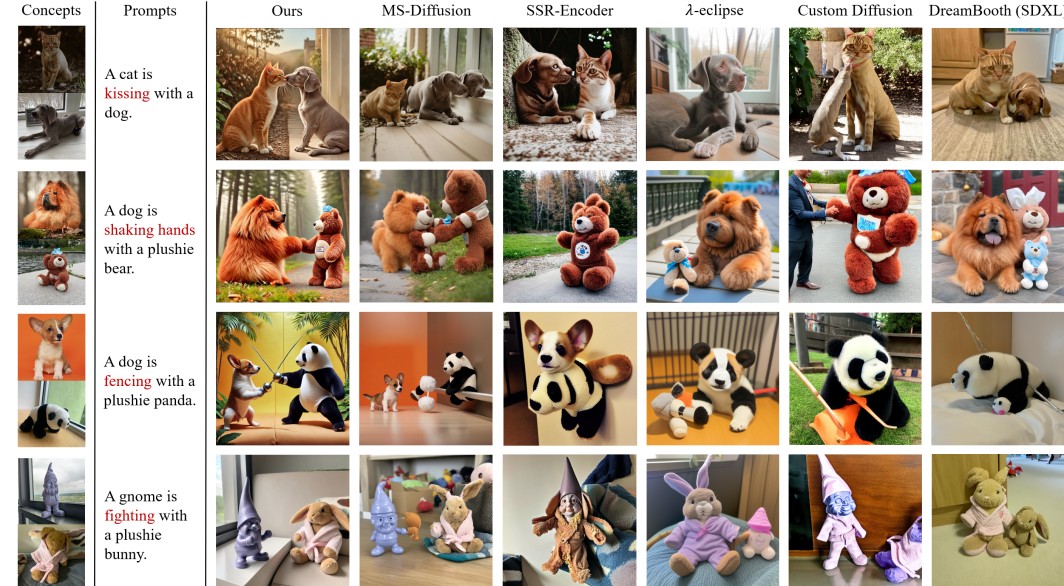

Figure 6: Multi-object comparison with training-based and tuning-based methods: Compared to our base model, our method shows a clear advantage in relation-aware generation and avoiding object confusion in overlapping scenarios.

Table 2: DreamBench.

| Method | Single-object | | |
|---|---|---|---|
| | CLIP-T | CLIP-I | DINO |
| DreamBooth (SDXL) | 31.2 | 81.5 | **69.2** |
| Custom Diffusion | 28.4 | 77.2 | 66.8 |
| Cones-V2 | 31.0 | 76.5 | 67.2 |
| ELITE | 29.8 | 77.4 | 62.5 |
| AnyDoor | 25.5 | **82.1** | 67.8 |
| SSR-Encoder | 30.8 | **82.1** | 61.2 |
| MS-Diffusion | 31.5 | 79.3 | 68.2 |
| Ours | **31.6** | 78.9 | 67.4 |

Table 3: M-CustomConcept101.

| Method | Multi-object | | |
|---|---|---|---|
| | CLIP-T | CLIP-I | DINO |
| DreamBooth (SDXL) | 29.5 | 67.4 | 49.2 |
| Custom Diffusion | 28.1 | 66.2 | 48.8 |
| Cones-V2 | 29.2 | 66.5 | 47.2 |
| AnyDoor | 20.2 | **72.1** | 51.2 |
| SSR-Encoder | **30.6** | 71.1 | 52.2 |
| λ-eclipse | 29.2 | 68.2 | 48.2 |
| MS-Diffusion | 28.0 | 70.2 | 51.2 |
| Ours | 29.4 | 71.4 | **52.3** |

Alignment, which evaluates how well the generated images align with the text prompts, using the CLIP image-text score, denoted as CLIP-T; and (3) Relation Alignment. We observed that nouns in the prompts can inflate CLIP-T scores, making them less accurate for evaluating relation generation. To address this, we extract relation predicate from the prompts using spaCy and calculate the CLIP image-text score using only the predicate, denoted as CLIP-R.

## 4.1 MAIN RESULTS

We evaluate our method through both quantitative and qualitative results in single- and multi-object cases. In single-object cases, we primarily focus on the alignment of the object's pose with the predicate relation in the text prompt, as well as the model's ability to preserve identity. In multi-object cases, in addition to relation generation and identity preservation, we also need to ensure there is no confusion between the multiple objects.

**Quantitative results.** We quantitatively compare our method with baseline models across three benchmarks: RelationBench, DreamBench, and multi-object cases in CustomConcept101. As shown in Tab. 1, our method demonstrates a clear advantage on RelationBench for single-object cases, particularly in the CLIP-T and CLIP-R metrics. However, it performs slightly lower in the CLIP-I metric, likely due to significant pose variations in the objects, which can negatively impact the CLIP-I score. For multi-object generation, our method surpasses other approaches across all evaluation metrics, which we attribute to effectively preventing object confusion, leading to a significant improvement in CLIP-I performance. Furthermore, as shown in Tab. 2, we evaluate single-object generation performance on DreamBench. Our method outperforms others in the CLIP-T metric while delivering competitive results in CLIP-I and DINO. To evaluate multi-object gener-

| Objects | Partner dance | Carry | Shake hands | Hug | Fight |
|---|---|---|---|---|---|

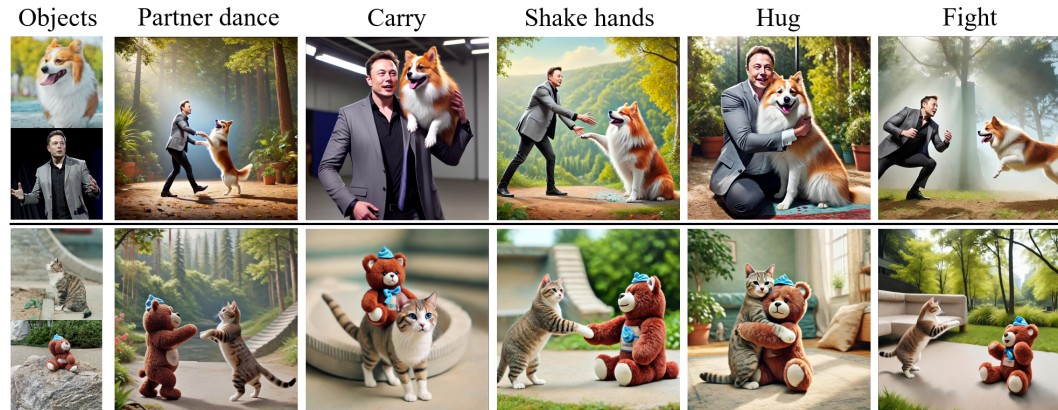

Figure 7: Additional results of relation-aware generation.

| Shake hands | Fight | Play chess | Carry | Hug |
|---|---|---|---|---|

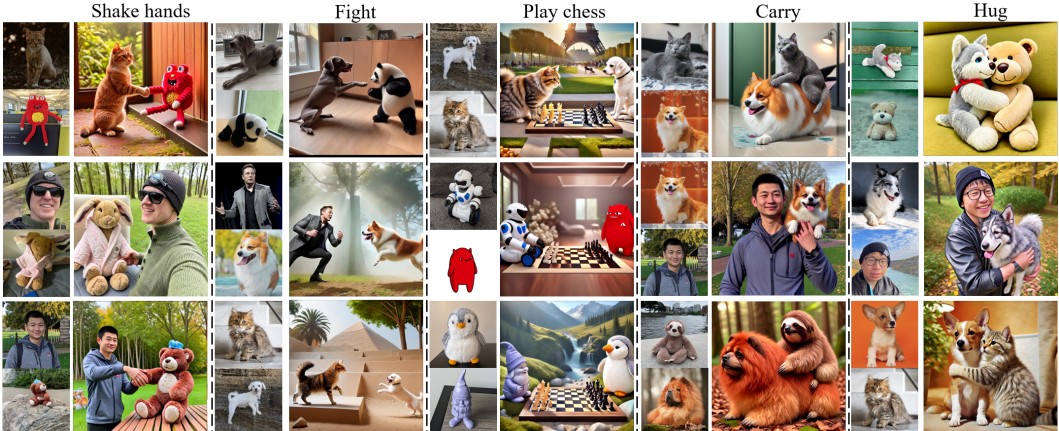

Figure 8: Additional results of relation-aware generation across a wide range of objects.

ation, we conduct experiments on multi-object cases from CustomConcept101, which we denote as M-CustomConcept101. As shown in Fig. 3, our model achieves the highest performance in the DINO score and ranks second in both the CLIP-T and CLIP-I metrics.

**Qualitative Comparison.** We conduct qualitative comparison experiments for relation-aware customized image generation on RelationBench. For single-object comparisons, as shown in Fig. 5, our generated images are more accurate than those produced by our base model, MS-Diffusion, and naturally reflect the predicate relations in the text prompts. Additional single-object comparison results are provided in the appendix (Fig. 12). In multi-object comparisons, Fig. 6 demonstrates that our method excels in generating multi-object relations, successfully avoiding object confusion while accurately representing the intended relations.

## 4.2 ABLATION STUDY AND ANALYSIS

**RelationLoRA Fine-tuning.** Instead of injecting LoRA layers and fine-tuning on our curated $(x, c_i, c_t)$, we fine-tune the text embedding $R^*$ using a set of relation-specific data $(x, c_t)$, where blank images are used as image prompts $c_i$. The visual comparison is presented in Fig. 4, and the quantitative results in Tab. 4 indicate that this method underperforms in both CLIP-T and CLIP-R scores. This may be due to the entanglement of identity and relation information in $x$, as well as the gap between the blank and real images, making it difficult to accurately generate the relation.

**Keypoint Matching Loss.** For the keypoint matching loss (KML), we omit it during fine-tuning. As shown in Fig. 9 a), including the KML leads to more accurate and natural relation-aware images when using the same random seed. Quantitative results, presented in Tab. 4, show a decline across all evaluation metrics when the KML is removed. Additionally, we experimented with different values for $\lambda$ and found that the best performance occurs at $\lambda = 1e - 3$.

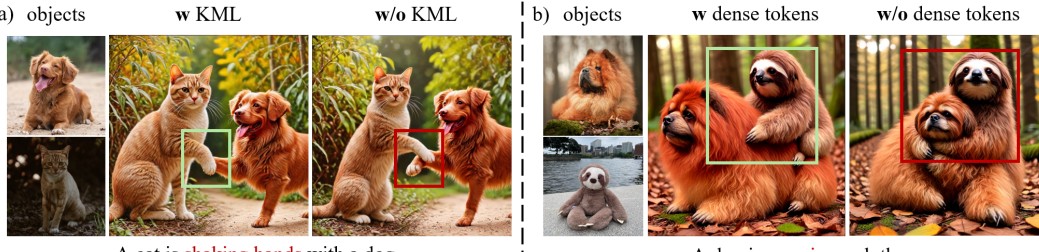

a) objects  **w** KML  **w/o** KML    b) objects  **w** dense tokens  **w/o** dense tokens

A cat is shaking hands with a dog.    A dog is carrying a sloth.

Figure 9: Ablation study of keypoint matching loss (KML) and local tokens: The images are generated using the same random seed. We use green and red boxes to highlight the main differences. The areas within the green boxes show the improvements with our proposed component.

Table 4: Ablation study on our proposed components

| Method | Single-object | | | | Multi-object | | | |
|---|---|---|---|---|---|---|---|---|
| | CLIP-T | CLIP-R | CLIP-I | DINO | CLIP-T | CLIP-R | CLIP-I | DINO |
| w/o Relation-aware Data | 28.4 | 16.9 | **78.2** | **64.4** | 26.2 | 16.8 | 75.3 | 59.7 |
| w/o Dense Token | 29.2 | 21.0 | 76.8 | 62.5 | 28.5 | 19.5 | 75.1 | 59.3 |
| w/o Keypoint Matching Loss | 28.3 | 17.9 | 77.8 | 62.1 | 27.4 | 18.2 | 75.2 | 61.2 |
| Full Model | **30.6** | **21.4** | 77.9 | 63.4 | **28.9** | **20.4** | **75.4** | **62.1** |

Table 5: Ablation study on Local Image Encoder's architecture.

| Model | Multi-object | | | |
|---|---|---|---|---|
| | CLIP-T | CLIP-R | CLIP-I | DINO |
| EVA-CLIP-L14 | 23.6 | 15.7 | 56.4 | 54.8 |
| CLIP-ViT-L14 | 22.9 | 14.8 | 58.3 | 52.7 |
| CLIP-ViT-bigG | **28.9** | **20.4** | **75.4** | **62.1** |

Table 6: Ablation study on $\lambda$, controlling relative weight of KML.

| Lambda | Multi-object | | | |
|---|---|---|---|---|
| | CLIP-T | CLIP-R | CLIP-I | DINO |
| $\lambda$=1e-2 | 27.5 | 19.3 | 72.6 | 59.6 |
| $\lambda$=1e-3 | **28.9** | **20.4** | **75.4** | **62.1** |
| $\lambda$=1e-4 | 26.8 | 18.1 | 73.9 | 60.4 |

Table 7: Ablation study on the number of local tokens (determined by partition size).

| Num Local Tokens | Multi-object | | | |
|---|---|---|---|---|
| | CLIP-T | CLIP-R | CLIP-I | DINO |
| N=2×2 | 28.3 | 19.4 | 75.3 | 60.1 |
| N=4×4 | **28.9** | **20.4** | **75.4** | **62.1** |
| N=8×8 | 27.9 | 18.1 | 74.1 | 59.4 |

Table 8: Ablation study on local token injection methods to ID extractor.

| Injection method | Multi-object | | | |
|---|---|---|---|---|
| | CLIP-T | CLIP-R | CLIP-I | DINO |
| Add | 25.4 | 18.5 | 71.0 | 56.9 |
| Linear Projection | 25.8 | 18.3 | 68.2 | 54.4 |
| Concatenate | **28.9** | **20.4** | **75.4** | **62.1** |

**Local Token Injection.** For local token injection, the visual comparison in Fig. 9 b) shows severe object confusion when local tokens are omitted. The quantitative results in Tab. 8 demonstrate that local tokens improve relation-aware customization. We conduct further ablation as shown in Tab. 7, Tab. 5, and Tab. 8. We experimented with varying numbers of local tokens injected into the ID extractor and found that 16 tokens yielded the best results across all evaluation metrics. Therefore, we use 16 local tokens in all experiments. Additionally, we swapped the Local Image Encoder and found that CLIP-ViT-bigG was the most compatible model. For the injection method, we note that simply concatenating local tokens with image tokens produced the best outcomes. Combining these methods ensures that our method maintains high quality relation-aware customized generation.

## 5 CONCLUSION

In this work, we introduce a challenging task: relation-aware customized image generation, which aims to generate objects that adhere to the relations specified in the text prompt while preserving the identities of user-provided images. To support this, we propose a data engine for generating high-quality fine-tuning data. Our method, RelationBooth, incorporating Keypoint Matching Loss and Local Token Injection, effectively captures relation information and generates natural relations between customized objects while mitigating object confusion. We also provide a fair comparison with other methods on a new benchmark, RelationBench. The extensive experiments on three benchmarks demonstrate the effectiveness of our method, highlighting its potential for interactive scenario generation, relation detection, etc.

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

# A APPENDIX

This appendix provides additional implementation details for the Local Image Encoder and baseline methods. In the second section, we offer a brief introduction to the RelationBench benchmark. We then present further experimental results to validate the effectiveness of our method across various scenarios. Following that, we provide an in-depth analysis of the contributions of the proposed components to overall performance. Lastly, we discuss the potential ethical, privacy, and fairness implications of this technology, highlighting its social responsibility in real-world applications.

## A.1 MORE IMPLEMENTATION DETAILS

**Local Image Encoder's Implementation Details.** To enhance region-language alignment of dense features, we employ self-distillation on CLIP-ViT-bigG. The training is conducted on the train2017 split of the COCO dataset for six epochs, using 8 A100 GPUs with a batch size of 2 per GPU. We apply the Adam optimizer with a learning rate of $1e-5$ and a weight decay of 0.1. The Local Image Encoder, containing 1.8B parameters, extracts local tokens that are concatenated with image tokens during fine-tuning and inference to improve the alignment of object appearance. This distillation process enhances the performance of our base model, MS-Diffusion, in multi-object generation tasks. However, significant object confusion still occurs during relation generation.

**Baselines' Implementation Details.** For tuning-based methods, we implement Textual Inversion, DreamBooth, and Custom Diffusion using their respective Diffusers versions, with learning rates and tuning steps aligned to those reported in the original papers. We utilize the official implementations and checkpoints for training-based methods, adjusting hyperparameters as needed during evaluation. Specifically, we set the scale to 0.6 in MS-Diffusion and sample 30 steps using the EulerDiscreteScheduler. For the SSR Encoder, we employ the UniPCMultistepScheduler, sampling 30 steps and adjusting the scale for each object to accommodate different cases. For $\lambda$-Eclipse, we apply the default settings of the official implementation without modification.

## A.2 RELATIONBENCH

In this section, we show the object and text prompt contained in our RelationBench in Fig. 10.

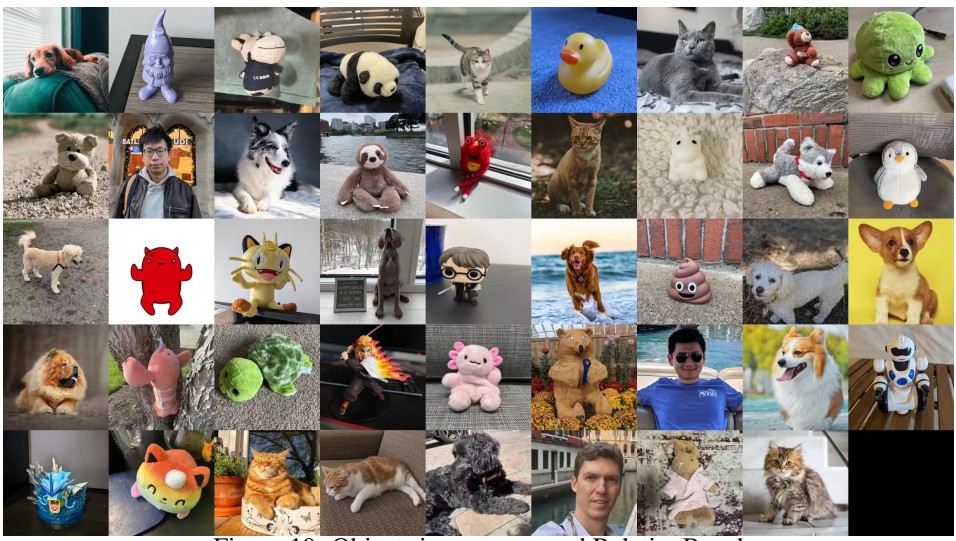

Figure 10: Objects in our proposed RelationBench

## A.3 MORE RESULTS OF RELATION-AWARE IMAGE CUSTOMIZATION

**More Results of Single-object relation-aware generation.** We show more qualitative results in Fig. 12. We compare our methods with both training-based and tuning-based methods.

Figure 11: Object category in RelationBench

Table 9: Text prompt in RelationBench.

| No. | Prompt |
| --- | --- |
| 1 | A { } is playing guitar on a park bench, serenading passersby. |
| 2 | A { } is playing piano in a grand hall filled with glowing chandeliers. |
| 3 | A { } is eating dinner in a bustling restaurant with soft jazz playing in the background. |
| 4 | A { } is dancing in the moonlight on a rooftop terrace with a stunning city view. |
| 5 | A { } is lifting weights in a modern gym, pushing through an intense workout. |
| 6 | A { } is reading a book by the fireplace on a rainy evening. |
| 7 | A { } is skiing down a steep slope in the Alps, with snowflakes falling gently. |
| 8 | A { } is sleeping peacefully in a hammock under the shade of a palm tree. |
| 9 | A { } is cooking lunch in an open-air kitchen overlooking a lush green valley. |
| 10 | A { } is singing on stage during a vibrant music festival with thousands of fans. |
| 11 | A { } is riding a bike along the scenic countryside road during sunset. |
| 12 | A { } is riding a horse through a dense forest, surrounded by nature. |
| 13 | A { } is riding a motorbike on a winding mountain road, feeling the wind rush by. |
| 14 | A { } is playing soccer on a sandy beach, with a vibrant sunset in the background. |
| 15 | A { } is playing chess with a { } in a quiet park under the shade of a tree. |
| 16 | A { } is partner dancing with a { } in a vintage ballroom with live music playing. |
| 17 | A { } is carrying a { } across a rushing river, carefully finding their steps. |
| 18 | A { } is fencing with a { } in an elegant arena, with spectators watching closely. |
| 19 | A { } shakes hands with a { } at a formal awards ceremony after receiving a prize. |
| 20 | A { } is kissing a { } in the rain under a large umbrella in a romantic city square. |
| 21 | A { } is playing basketball with a { } on a street court. |
| 22 | A { } is wrestling with a { } in a championship ring. |
| 23 | A { } is hugging a { } in front of the mountain. |
| 24 | A { } is fighting with a { } in a garden. |
| 25 | A { } is sitting back to back with a { } on a hilltop. |

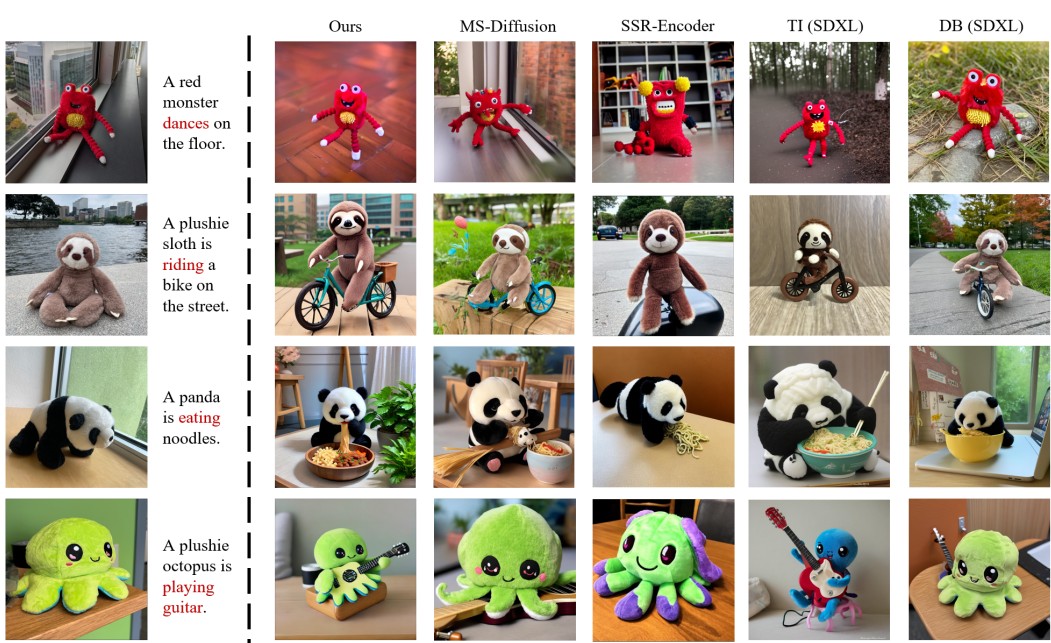

Figure 12: Single-object comparison. TI and DB indicate Textual Inversion and DreamBooth, respectively. Our methods achieve the best balance between relation generation and identity preservation.

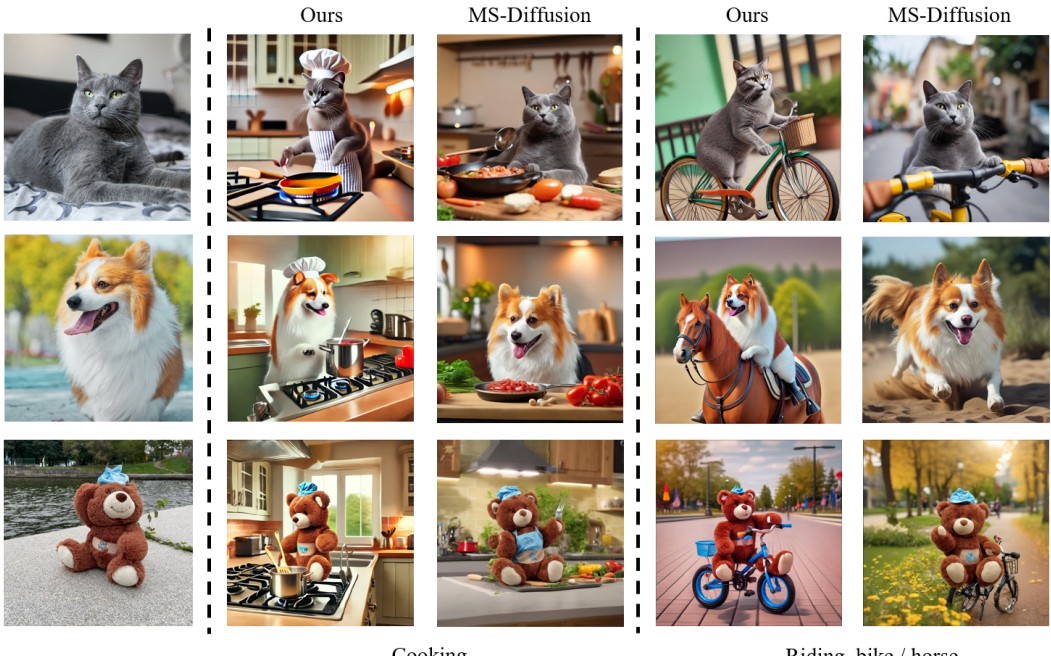

Figure 13: Single-object comparison with our base model MS-Diffusion

**More Results of Multi-object relation-aware generation.** In this section, we present additional results on multi-object relation-aware generation. In addition to implementing our RelationBooth on MS-Diffusion, we also use SDXL as a base model. As shown in Fig. 15, our RelationBooth enhances SDXL's ability to generate images that accurately depict object relations. Compared to ReVersion (Huang et al., 2023), our method generates more precise relations, without object confusion or missing objects, demonstrating the strong performance of RelationBooth in the Relation Generation task.

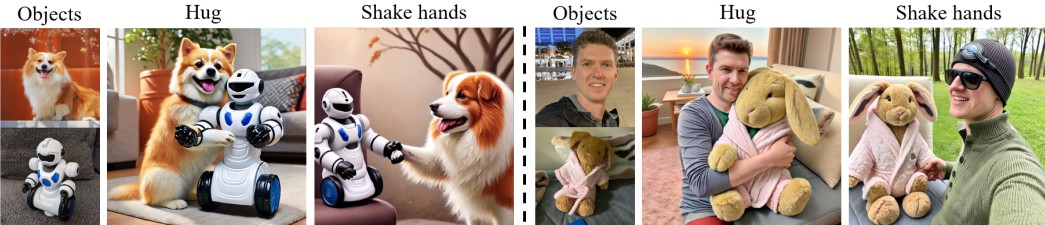

Figure 14: Multi-object relation-aware image customization results of pet, toy, plushie, and person.

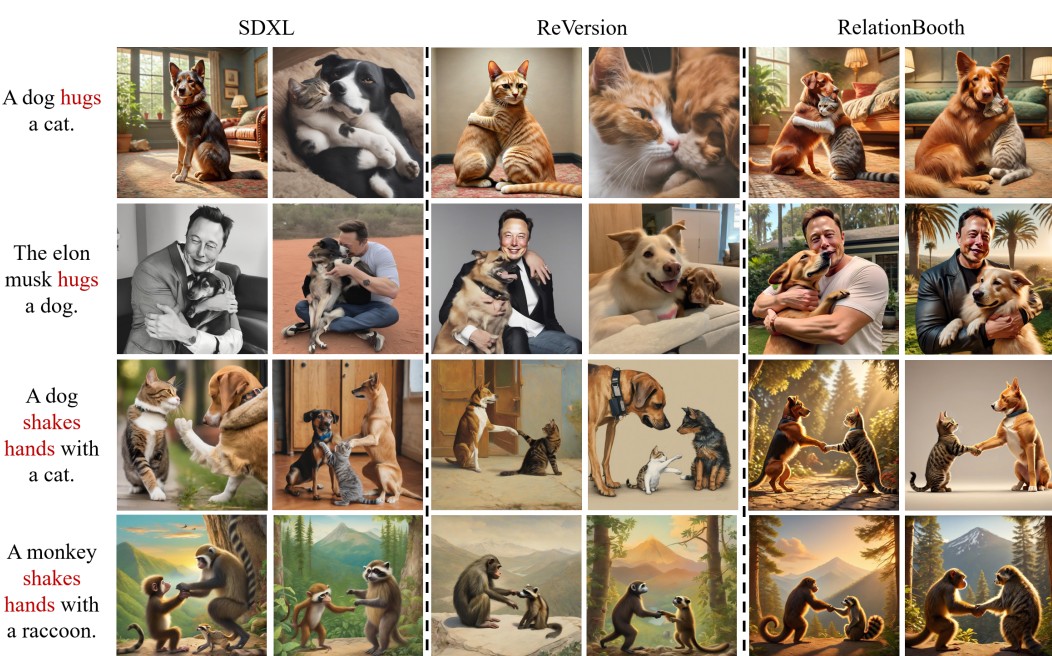

Figure 15: Our RelationBooth is compatible with SDXL to address Relation Inversion task.

**Failure cases.** As shown in Fig. 16, we present three typical failure cases from our experimental results. The first involves unreasonable relation generation requests, such as asking a plushie octopus, which lacks limbs, to 'shake hands.' In response, our model generates additional arms for the plushie octopus, leading to a mismatched appearance. The second issue is the unnaturalness of some generated relations, such as a duck failing to make contact with a cat as it should. The final failure case is object confusion at the interaction point, which is a common challenge across all multi-object generation models.

### A.4 ADDITIONAL ANALYSIS ON OUR PROPOSED COMPONENTS

**Our well-curated Data.** Our curated data are generated by DALLE-3 (Betker et al., 2023), which can effectively preserve the identity of common categories, such as animals, through its multi-turn dialogue capability. While our fine-tuning dataset consists solely of animals, our method focuses primarily on the relation information in the ground truth by disentangling identity using an off-the-shelf identity extractor. Therefore, the specific categories in the fine-tuning data are not the most critical factor for relation learning.

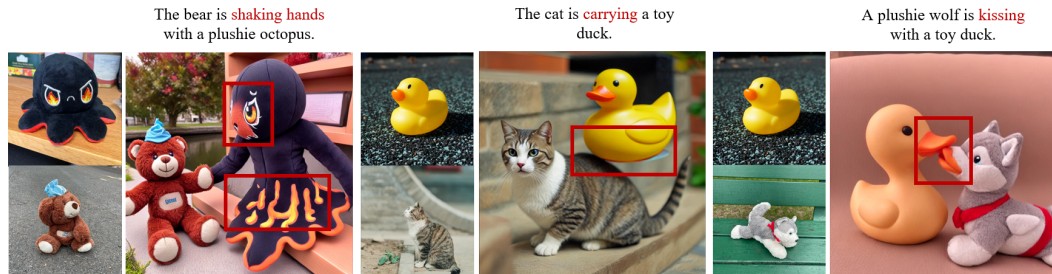

Figure 16: Failure cases of our RelationBooth.

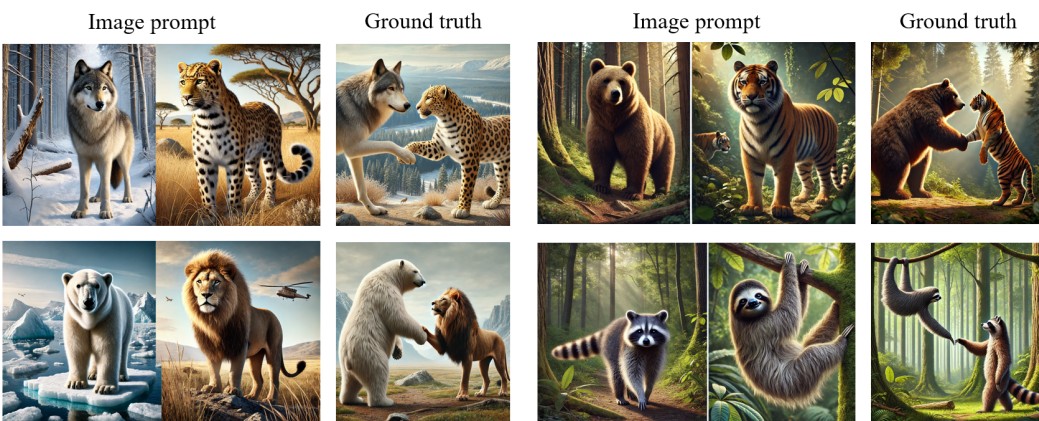

Figure 17: Our fine-tuning dataset as an example.

**Keypoint Matching Loss.** We use X-Pose as our keypoint detector due to its open-vocabulary detection capabilities. The keypoint matching loss (KML) facilitates relation generation by explicitly guiding the model's pose manipulation, resulting in more accurate pose generation. As shown in Fig. 19, the cat's arm crosses the dog's body, accurately depicting the 'hug' relation.

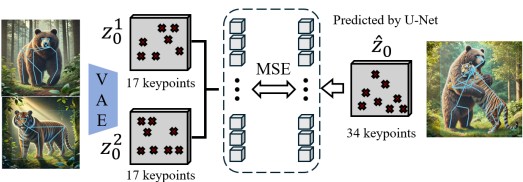

Figure 18: Illustration of Keypoint Matching Loss

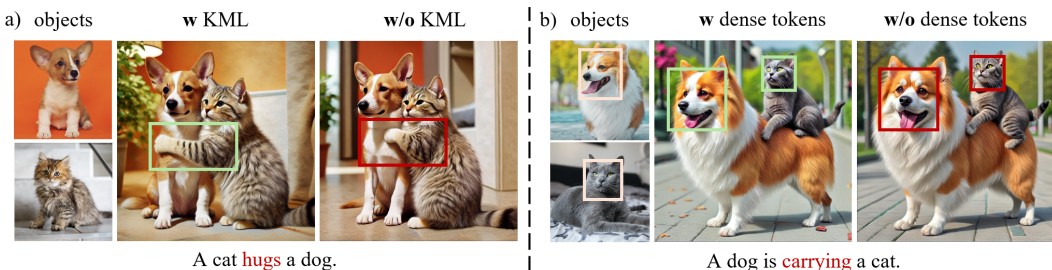

Figure 19: Additional ablation studies on KML and Local Tokens.

**Local Tokens Visualization.** To understand why local features enhance relation-aware generation, we adopt Principal Component Analysis (PCA) to project the dense feature more compactly. As shown in Fig. 20, dense features provide more fine-grained information than CLIP image tokens, aiding in distinguishing between different objects during the generation process and helping to avoid

object confusion, especially in cases of heavy overlap. Moreover, it can facilitate object appearance alignment.

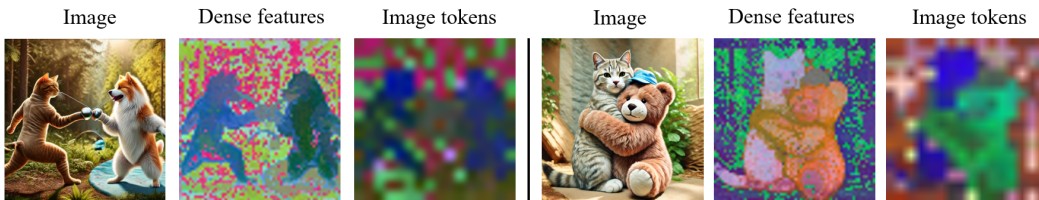

Figure 20: Visualization of Dense feature and Image tokens by Principal Component Analysis (PCA).

### A.5 SOCIAL COMPACT

**Positive societal impacts.** Relation-aware image customization enables users to generate images that not only contain customized objects but also capture their meaningful relationships. This opens up new opportunities for creative professionals, such as designers, advertisers, and educators, to communicate complex ideas visually with greater precision and flexibility. It has the potential to streamline content creation in diverse fields, from personalized marketing to educational tools, making high-quality, contextually rich imagery accessible without the need for extensive resources.

**Potential negative societal impacts.** The ability to generate customized images that involve specific relationships between objects could be misused to fabricate misleading or harmful visual narratives, including false representations of events or manipulative visual content in political or social contexts. Additionally, if the models are trained on biased data, they may reinforce existing societal biases, marginalizing certain groups or distorting the accuracy of represented relationships.

**Mitigation strategies.** To reduce misuse, ethical guidelines should be established to govern the responsible development and application of this technology. Promoting transparency about generated content and integrating fairness and diversity considerations into dataset selection are key strategies for mitigating potential harms.

