# OpenReview forum: "RelationBooth: Towards Relation-Aware Customized Object Generation"
_ICLR.cc/2025/Conference — ICLR 2025 Conference Withdrawn Submission_

### Official Review · Reviewer_9QXH · 2024-11-01

**Soundness:** 3
**Presentation:** 3
**Contribution:** 2
**Rating:** 5
**Confidence:** 5

**Summary:**

This paper explores a new task called relation-aware customized image generation. The task involves maintaining multiple identities while adhering to the relationships specified in text prompts. This paper constructed a new dataset for this task, comprising relation-specific images, independent object images containing identity information, and text prompts to guide relation generation. The proposed method for this task, Relation Booth, incorporates two key modules: keypoint matching loss that guides the model in adjusting object poses closely tied to their relationships and object-related local tokens to enhance identity preservation.

**Strengths:**

1. The paper presents a strong motivation by identifying shortcomings in existing predicate relationship generation methods, leading to this new task's proposal.
2. It introduces an automated dataset generation method, enhancing the study's practical application.

**Weaknesses:**

1. The experimental results show optimal performance only on the self-constructed dataset, with suboptimal performance on the other two datasets, failing to demonstrate the method's validity convincingly.
2. In section 4.2 RelationLoRA Finetuning, Using solely black images when finetuning text embedding as image prompts may lead to unfair comparisons.
3. The model functions as a plug-and-play module, relying on MS-Diffusion as its baseline, which limits its originality.

**Questions:**

1. In section 3.4.2, the KML loss uses C_kp^{C_i} as the supervisory signal. Why not utilize the interleaved skeletons from c_kp^x to align with interactive actions?
2. Moving the Local Tokens Visualization from the appendix to the paper's main body is recommended and includes a more in-depth analysis.
3. Conduct fair experimental validations of the method's effectiveness on additional publicly available datasets. For instance, select images of people from datasets such as LAION-400M for experimentation.

---

### Official Review · Reviewer_1GyU · 2024-11-02

**Soundness:** 2
**Presentation:** 2
**Contribution:** 2
**Rating:** 3
**Confidence:** 5

**Summary:**

This paper introduces RelationBooth, a framework designed for relation-aware object customization. This framework includes a data engine designed to curate a dataset comprising independent customized object images, relation-specific images, and text prompts to guide relation generation. Then it finetune the diffusion model on the dataset collected by the proposed data engine with keypoint matching loss and local tokens to improve relation accuracy and prevent object confusion. Experimental results indicate the framework's potential for enabling precise multi-object interactions in personalized content creation.

**Strengths:**

1. The relation-aware image customization, especially for the multi-objects customization with interaction, is useful in real world application.
2. RelationBooth effectively preserves object identities.
3. The relations among objects is well-aligned with the text prompt in the visualizations.

**Weaknesses:**

1. The writing should be improved. The illustrations require improvements for better comprehension, and grammatical errors, notably in L247-L249, should be addressed.
2. The novelty of this paper is limited. Customizing objects using both image and text prompts is a concept that has been previously explored [1,3,4]. Additionally, prior research has delved into multi-object customization [2,5,6] and action customization [4,7,8,9]. The authors should highlight their unique contributions beyond combining similar tasks in this area.
3、The illustration and analysis of the training dataset is limited. More visual examples of the training samples should be included in the paper.
4. The analysis of the limitation, e.g., generalization of different relations and objects should be included.

[1] IP-Adapter: Text Compatible Image Prompt Adapter for Text-to-Image Diffusion Models
[2] Mix-of-Show: Decentralized Low-Rank Adaptation for Multi-Concept Customization of Diffusion Models
[3] Tuning-Free Image Customization with Image and Text Guidance
[4] Learning Disentangled Identifiers for Action-Customized Text-to-Image Generation
[5] OMG: Occlusion-friendly Personalized Multi-concept Generation in Diffusion Models
[6] FastComposer: Tuning-Free Multi-Subject Image Generation with Localized Attention
[7] MotionDirector: Motion Customization of Text-to-Video Diffusion Models
[8] ReVersion: Diffusion-Based Relation Inversion from Images
[9] DreamVideo: Composing Your Dream Videos with Customized Subject and Motion

**Questions:**

1. If I understand correctly, the data engine introduced in Sec 3.3 will generate triples D = （x, c_i, c_i） optimize the LoRAs injected into the text-conditioned cross-attention (TCA) layers. How do these LoRAs relate to object customization tasks? Are they intended for injected into an image-customized model?

2. According to Sec 3.4.1 and Equ 4, the parallel cross-attention layers are the same with IP-Adapter. However, IP-Adapter trains the newly added image-conditioned cross-attention layers while the pretrained text-to-image model is frozen. I understand that in RelationBooth, fine-tuning LoRAs attached to TCA is used for relations learning. However, given that the initial stable diffusion model solely comprises text-conditioned cross-attention layers, why is there no requirement for training the newly incorporated image-conditioned layers?
[1] IP-Adapter: Text Compatible Image Prompt Adapter for Text-to-Image Diffusion Models

3. In Sec 3.1, the number of image prompts is set to 2, which indicates that RelationBooth can only handle two-object interaction generation. So my questions is, what the results be like when N=3. In real life, many relationships involve more than two objects, including fighting and hugging.

4. In Sec 3.4.3, what operation follows after obtaining h_{tmp}?  How is h_{clip} actually modified? It appears that the authors may have typos or symbols confusion in Equ.8.

5. When conducting experiments, do all methods use the same SD versions? As far as I know, comparison methods like Custom Diffusion, ELITE, and BLIP-Diffusion are all based on SD1.5. However,  RelationBooth is implemented based on SDXL. Different SD version have different image generation capabilities. Have the author considered this issue in Table 1?

---

### Official Review · Reviewer_jf4K · 2024-11-04

**Soundness:** 3
**Presentation:** 3
**Contribution:** 3
**Rating:** 6
**Confidence:** 4

**Summary:**

In this paper, the authors proposed a object generation method based on object relation awareness. The proposed work has two main components, keypoint matching loss and local token injection. The experiments show the effectiveness of the proposed work.

**Strengths:**

+ The problem is interesting.
+ The proposed framework is straightforward and easy to follow.
+ The authors conducted a lot of ablation studies.
+ The visualization showcases the strength of the proposed method.

**Weaknesses:**

+ I have concerns regarding the performance of the proposed work. As shown in Tables 1, 2, and 3, the proposed work is outperformed by baselines in certain metrics.
+ The ablation study does not show the strength of having a full model. As shown in Table 4, w/o Relation-aware Data achieves a better performance in terms of CLIP-I and DINO.
+ There are still some image synthesis errors such as 6 fingers on one hand.

**Questions:**

I have questions regarding the performance of the proposed work, the ablation study, and some image synthesis errors.

---

### Official Review · Reviewer_NR6i · 2024-11-04

**Soundness:** 2
**Presentation:** 3
**Contribution:** 2
**Rating:** 3
**Confidence:** 5

**Summary:**

This paper proposes RelationBooth, focusing on relation-aware customized image generation. Based on a zero-shot multi-subject personalized model, the authors introduce a key point loss and local tokens to solve the relation failure in the SOTAs.

**Strengths:**

1. As claimed by the authors, they are the first to address the generation of relation-aware customized images, which is an important issue in customized image generation.
2. The experimental results show that they get fancy performance compared to SOTA customized methods.

**Weaknesses:**

1. The novelty of this paper is somehow low. Although the authors claim to be the first customized method to solve the relation-aware issue,  this issue has already been explored in the normal diffusion models, as the authors mentioned in the related work. I wonder if the relation-aware methods except RI indicated in Fig. 4 can be used with customized models like IP-Adapter [1], SSR-Encoder [2], and MS-Diffusion [3]. If so, I suggest the authors add detailed experiments about the difference between RelationBooth and other relation-aware image generation methods.
2. The contribution is weak since the proposed model is fine-tuned on MS-Diffusion using LoRAs. The design of local tokens is also an application of other research.
3. As indicated by the quantitative results in Tab. 1, it seems that the image similarity is sacrificed to increase the text similarity. This is also reflected by some qualitative examples in Fig. 6 and Fig. 8.

[1] Ye, Hu, et al. "Ip-adapter: Text compatible image prompt adapter for text-to-image diffusion models." arXiv preprint arXiv:2308.06721 (2023).

[2] Zhang, Yuxuan, et al. "Ssr-encoder: Encoding selective subject representation for subject-driven generation." Proceedings of the IEEE/CVF Conference on Computer Vision and Pattern Recognition. 2024.

[3] Wang, X., et al. "MS-Diffusion: Multi-subject Zero-shot Image Personalization with Layout Guidance." arXiv preprint arXiv:2406.07209 (2024).

**Questions:**

Please see the weakness.

---

### Note · Authors · 2024-11-13

I have read and agree with the venue's withdrawal policy on behalf of myself and my co-authors.